

# How good are hydrological models for gap-filling
# streamflow data?
Yongqiang Zhang[1*], David Post[1]
[1] CSIRO Land and Water, GPO Box 1700, ACTON 2601, Canberra, Australia
[*]Corresponding author: Yongqiang Zhang
Email: yongqiang.zhang@csiro.au; yongqiang.zhang2014@gmail.com
Address: CSIRO Land and Water, Clunies Ross Street, Canberra 2601, Australia
Tel.: +61 2 6246 5761





**Key Points:**
• Gap-filling of streamflow data performs well when the missing rate is less than 10%
• Small number of catchments showing large trend bias when the missing rate is up to

22 20%

• Poor gap-filling occurring in some wet catchments even with reasonable model
calibration


**Abstract.** Gap-filling streamflow data is a critical step for most hydrological studies, such
as streamflow trend, flood and drought analysis and hydrological response variable estimates
and predictions. However, there is lack of quantitative evaluation of the gap-filled data
accuracy in most hydrological studies. Here we show that when the missing rate is less than
10%, the gap-filled streamflow data obtained using calibrated hydrological models perform
almost as same as the benchmark data (less than 1% missing) for estimating annual trends for
217 unregulated catchments widely spread in Australia. Furthermore, the relative streamflow
trend bias caused by the gap-filling is not very large in very dry catchments where the
hydrological model calibration is normally poor. Our results clearly demonstrate that the gap-
filling using hydrological modelling has little impact on the estimation of annual streamflow
and its trends.
**Keywords:** streamflow, data, gap-filled, hydrological model, trend




# 1 Introduction

Streamflow is channel runoff, i.e. the flow of water in streams and rivers and accumulated

from surface runoff from land surface and groundwater recharge. It is one of the major water

balance components in a catchment where precipitation is partially stored in surface water,

soil and groundwater stores, and the rest is partitioned into two fluxes: evapotranspiration and

streamflow. It is almost impossible to measure evapotranspiration dynamics at a catchment

scale. In contrast, streamflow time series can be easily measured at a catchment outlet.

Therefore, streamflow data becomes a fundamental dataset underpinning hydrological

studies. Without such a dataset, it is hard to understand catchment hydrological processes

under climate change and non-stationarity(Dai et al., 2009; Gedney et al., 2006a; Ukkola et

al., 2015; Zhang et al., 2016b).

Unfortunately, streamflow data are not always continuously available and most gauges suffer

from streamflow data missing issues(Dai et al., 2009). Often, the missing rate is important

when selecting streamflow gauges, especially when the data is used for annual trend analysis.

To choose qualified catchments, researchers often set up a threshold for the missing ratio, for

instance 1%(Petrone et al., 2010), 5%(Ukkola et al., 2015), 10%(Déry et al., 2009), 15%

(Liu and Zhang, 2017), and 20%(Lopes et al., 2016). Only those gauges with missing rate

less than a particular threshold are selected, and the rest are excluded for further analysis

because of high missing rates.

There are many methods used for gap-filling the missing data, including interpolation from

nearby gauges(Lopes et al., 2016), statistical methods(Gedney et al., 2006b), and

hydrological modelling(Dai et al., 2009). Among them, the hydrological modelling method

is widely used since it fully considers the spatial heterogeneity and temporal variability of

climate forcing data, and can achieve sufficient simulations when it is calibrated against a





small number of observations (Rojas-Serna et al., 2016; Seibert and Beven, 2009). This is
particularly important in Australia where hydrological modelling is a major tool for
simulating continuous streamflow at a catchment scale. More recently, the Australian Bureau
of Meteorology used a hydrological model –GR4J– to infill missing daily streamflow data for
222 Hydrologic Reference Stations (http://www.bom.gov.au/water/hrs/about.shtml). The gap-
filled streamflow data are then used for trend analysis and providing hydrological information
to all users.
One major concern for the hydrology community is to understand how reliable the gap-filled
data is. Unfortunately there are no studies in the literature to comprehensively evaluate the
reliability and accuracy of the gap-filled data that are influenced by different thresholds and
by data missing patterns. Our study aims to provide a framework to evaluate the annual
trends and annual variables obtained from gap-filled streamflow data using two hydrological
models (GR4J and SIMHYD) together with a large streamflow dataset available across the
Australian Continent(Zhang et al., 2013). This can guide researchers to more sensibly define
a threshold for catchment selection and hydrological analysis.

## 2   Data and Methods

### 2.1 Data

We obtained daily streamflow data set from 780 unregulated catchments widely spread across
Australia(Zhang et al., 2013). The dataset has undergone strict quality assurance and quality
control, including quality codes check and spike (i.e. outlier points) control, and covered the
period from 1975 to 2012. This dataset has been used by modellers for various hydrological
modelling and extreme-event studies(Li and Zhang, 2017; Liu and Zhang, 2017; Ukkola et
al., 2016; Yang et al., 2017). The missing rate for the pre-1980 and post-2010 periods were
high. To meet our study requirement, we selected 217 catchments with a data missing rate



less than 1% for the period 1981-2010 and the streamflow data for the 217 catchments are
regarded as 'benchmark' data (Figure 1). Out of the 780 catchments there are 146, 91, and 61
with the missing rate of 1-5%, 5-10%, and 10-20% during 1981-2010, respectively (Figure
1), and these catchments account for 38% of total available catchments. Table 1 summarises
major catchment attributes for the 217 selected catchments.

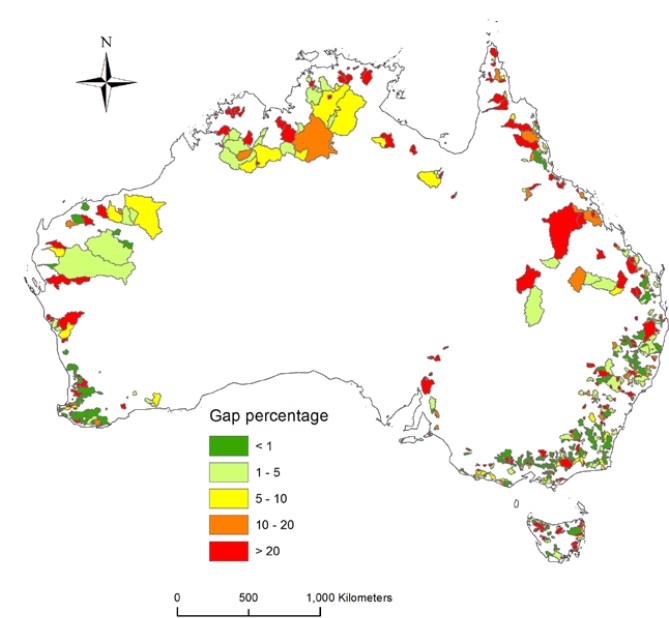


**Fig. 1.** The 780 unregulated catchments grouped by different streamflow data gaps for the
period of 1981-2010.

**Table 1.** Major catchment attributes for the 217 catchments

| Attribute | Definition | Unit | Min | 2.5th | 25th | Median | 75th | 97.5th | Max |
|---|---|---|---|---|---|---|---|---|---|
| Area | Catchment area | km² | 53 | 70 | 180 | 392 | 844 | 4562 | 72902 |
| Elevation | Catchment average elevation above sea level | m | 46 | 100 | 278 | 449 | 753 | 1194 | 1351 |
| Slope | Catchment mean slope | Degrees | 0.3 | 0.6 | 2.0 | 3.9 | 7.7 | 12.0 | 13.6 |
| P | Mean annual precipitation | mm/year | 256 | 371 | 703 | 853 | 1107 | 1966 | 2473 |





| | | | 906 | 968 | 1149 | 1235 | 1408 | 1791 | 1892 |
|---|---|---|---|---|---|---|---|---|---|
| $ET_p$ | Mean annual potential evapotranspiration | mm/year | | | | | | | |
| AI | Aridity index | - | 0.38 | 0.55 | 1.11 | 1.44 | 1.89 | 4.75 | 6.47 |
| Forest ratio | Ratio of forest to all land cover types | - | 0.02 | 0.06 | 0.39 | 0.55 | 0.67 | 0.83 | 0.90 |


Out of the 217 catchments, about half of the catchments showed a significant decreasing
trend, 37% showing non-significant decreasing trend, and 13% showing non-significant
increasing trend (Figure 2), detected using Mann-Kendall trend analysis (see 2.3). This is
because Australia experienced the Millennium drought over the period 2001-2009, which
caused a dramatic streamflow reduction in this period(van Dijk et al., 2013). Trend analysis
for the 217 catchments is explained in Section 2.3 and trend results are summarised in
Section 3.
Out of the 217 catchments, about 46% of catchments have no missing data in 1981-2010,
12% with the missing rate <0.1%, 22% with the missing rate 0.1-0.5% and 20% with the
missing rate of 0.5-1% (Figure 2).



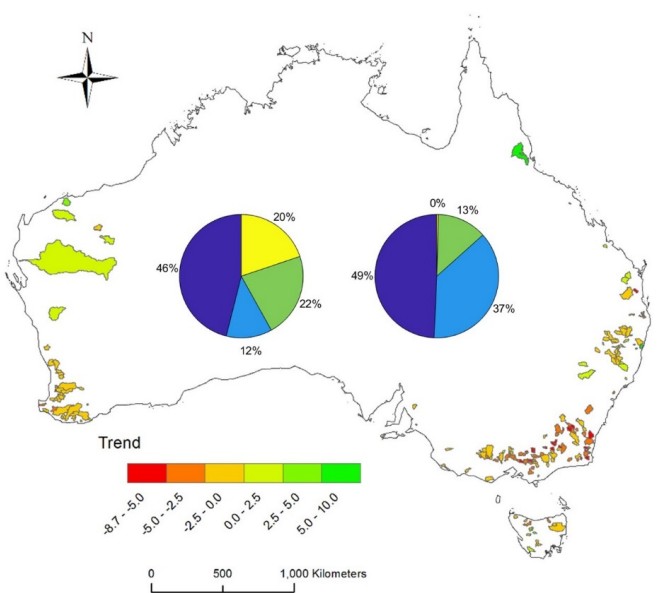


**Fig. 2.** Trends and streamflow data summary for the 217 catchments used in this study. Trend

in annual streamflow is with a unit of mm/year/year. Left pie indicates the catchment

percentage with different missing rates (dark blue with missing rate of 0%, navy blue with

missing rate of 0-0.1%, green with missing rate of 0.1-0.5%, yellow with missing rate of 0.5-

1.0%); right pie indicates the catchment percentage with different trends (dark blue with

significant ($p \leq 0.05$) decreasing trend, navy blue with non-significant ($p > 0.05$) decreasing

trend, green with non-significant ($p > 0.05$) increasing trend, and yellow with significant ($p \leq$

0.05) increasing trend).

To drive the two hydrological models, we obtained daily meteorological time series

(including minimum temperature, maximum temperature, incoming solar radiation, actual

vapour pressure and precipitation) from 1975 to 2012 at 0.05° (~5 km) grid resolution from

the SILO Data Drill of the Queensland Department of Natural Resources and Water

(www.nrw.gov.au/silo). The data quality is reasonably good, indicated by the mean absolute





error for maximum daily air temperature, minimum daily air temperature, vapour pressure,
and precipitation at 1.0°C, 1.4°C, 0.15 kPa and 0.40 mm/day(Jeffrey et al., 2001).
**2.2 Gap-filling experiments**
For thoroughly investigating the potential impacts of infilled streamflow data on annual trend
accuracy, we conducted three groups of experiments to test how the missing rates at 5%, 10%
and 20% impact on streamflow trends. We followed three steps for each missing rate of
experiments:
*1. Missing patterns were obtained using actual streamflow data.* We selected consecutive
missing day pattern from actual data from the 780 catchments. For 5% group of missing rate
experiments, we selected 44 catchments with missing rates in 4-6%; for 10% group of
missing rate experiments, we selected 39 catchment with missing rate in 8-12%; for 20%
group of missing rate experiments, we selected 22 catchments with missing rate in 18-22%.
Figure 3 shows the probability distribution of consecutive missing days from each group of
catchments, which is skewed toward the low end. We therefore used the two-parameter
Gamma distribution to simulate probability distribution of consecutive missing days (Figure
3). The Gamma distribution is expressed as
$$X : \Gamma(k,\theta) = Gamma(k,\theta),\tag{1}$$

where X is the consecutive missing days number, *k* is shape parameter, and $\theta$ is scale
parameter. The corresponding probability density function in the shape-scale
parameterization is
$$f(x;k,\theta) = \frac{1}{\Gamma(k)\theta^k} x^{k-1} e^{-\frac{x}{\theta}},\tag{2}$$

where $\Gamma(k)$ is the gamma function.



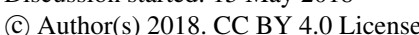



140

**Fig. 3.** Missing patterns for three groups of catchments with missing rates 4-6%, 8-12%, 18-

22% that represent 5%, 10% and 20% missing rates, respectively.

As seen from Figure 3, the two parameters are stable under the three groups of catchments.

The $k$ parameter varies from 0.63 to 0.87 and the $\theta$ parameter changes from 62 to 81. It is

noted that we removed all times when the number of consecutive missing days was > 365.

We did that for a number of reasons. Firstly, gap-filling an entire year of missing data would

likely impact annual trends. Secondly, the focus of this paper is on gap-filling short periods

of missing data to be able to include more catchments in streamflow analyses. Thirdly,

removing all periods of greater than 365 days allowed us to better fit a gamma distribution to

the number of missing days.

*2. Generating random consecutive missing day numbers using random number generator*

*(sampling without replacement) based on the Gamma distribution.* The random number

generator was repeated 100 times to ensure the selected samples cover a wide range of

streamflow time series.

*3. Gap-filling streamflow data.* The selected days were treated as 'missing' data and the

unselected data were used for hydrological model calibration. The 'missing' data were then

gap-filled using the simulated streamflow from the calibrated GR4J and SIMHYD models,

respectively.



For consistent interpretation thereafter, the benchmark streamflow data is regarded as
'observed' and the experiment ones as 'filled' ones. For each of the three experiments, there
are 100 x 217 (21,700) 'missing' time series, with 100 representing sample times using the
random number generator and 217 representing the number of catchments.
**2.3 Trend analysis**
We used the Mann–Kendall Tau-b non-parametric test including Sen's slope method(Burn
and Elnur, 2002) for annual streamflow trend analysis and significance testing for all the
three groups of experiments and benchmark data.
We used the following equation to quantify the trend bias*:*

174                                $$B_t = T_{filled} - T_{obs},$$                                (3)

where $B_t$ is the bias in annual streamflow trend (mm/year/year), $T_{filled}$ is annual trend for gap-
filled streamflow (mm/year/year), $T_{obs}$ is annual trend in observed streamflow
(mm/year/year). It measures the trend error between the infilled and observed runoff trends
with $B_t \approx 0$, which indicates that the trend in observed annual runoff is almost the same as
that in the infilled annual runoff.
We also defined relative trend bias ($P_{Bt}$) as

181                        $$P_{B_t} = \frac{T_{filled} - T_{obs}}{T_{obs}} \times 100 \ ,$$                        (4)


**2.4 Hydrological models**
Two widely used hydrological models SIMHYD and GR4J (Chiew et al., 2002; Chiew et al.,
2010; Li et al., 2014; Oudin et al., 2008; Perrin et al., 2003; Zhang and Chiew, 2009; Zhang
et al., 2016a) were used to infill daily 'missing' streamflow. Both models require daily





precipitation and daily potential evaporation (Priestley and Taylor, 1972) as model inputs,
and model outputs are daily streamflow at each gauge. The daily inputs of the maximum and
minimum temperatures, incoming solar radiation, and vapour pressure data were used to
calculate the Priestley–Taylor daily potential evaporation.
The two models were calibrated using a global optimiser: genetic algorithm (The
MathWorks, 2006) at each catchment, with the first six years (i.e., 1975–1980) for spin up
and remainder (1981 to 2010) for modelling experiments. Since this study mainly evaluates
the trends obtained using the gap-filled streamflow from hydrological modelling, it is crucial
to predict high flow and mean flow as accurate as possible. To this end, the model calibration
was to minimize the following objective function ($F$) (Viney et al., 2009; Zhang et al.,
2016b):

$$F = (1 - NSE) + 5\left|\ln(1 + B)\right|^{2.5}, \tag{5}$$


$$B = \frac{\sum\limits_{i=1}^{N} Q_{sim,i} - \sum\limits_{i=1}^{N} Q_{obs,i}}{\sum\limits_{i=1}^{N} Q_{obs,i}}, \tag{6}$$



where $NSE$ is the Nash-Sutcliffe-Efficiency of daily streamflow, $B$ is the model bias, $Q_{sim}$ and
$Q_{obs}$ are the simulated and observed daily runoff, $i$ is the $i$th day, $N$ is the total number of days
sampled. The $NSE$ gives higher streamflow more weight, and varies between $-\infty$ to 1 with
$NSE > 0.6$ indicating a good agreement (Zhang and Chiew, 2009). The $B$ measures water
balance error between the observed and modelled daily streamflow, with $B = 0$ indicating that
the average of modelled daily streamflow is the same as the average of observed daily
streamflow.
For each catchment, GR4J and SIMHYD were calibrated using benchmark data and 100 time
series of streamflow data with 'missing' data (see Section 2.2), respectively. For benchmark
data without any missing data (46% catchments) there are no gap-filling required; for the



benchmark data with missing rate less than 1%, the calibrated continuous streamflow data
were used to fill the gaps. For the 'missing' experiments, the calibrated continuous
streamflow data for each 'missing' replicate were used to infill the artificially-made 'missing'
data. Table 2 summarises the model calibrations carried out for benchmark and each
experiment. Finally, there were 1,302,434 model calibrations and 1,302,000 times of gap-
filling carried out. Finally, the trends estimated from benchmark were used to evaluate those
obtained from the 'missing' experiments.
**Table 2.** Summary of model calibration number carried out for benchmark and data 'missing'
experiments

| Model | Benchmark | 5% missing | 10% missing | 20% missing | Sum |
|---|---|---|---|---|---|
| GR4J | 217 | 217,000 | 217,000 | 217,000 | 651,217 |
| SIMHYD | 217 | 217,000 | 217,000 | 217,000 | 651,217 |
| Sum | 434 | 434,000 | 434,000 | 434,000 | 1,302,434 |


## 3 Results

The gap-filled data from the two hydrological models were evaluated against the benchmark
data. Figures 4 and 5 summarise the performance of the gap-filled data for estimating annual
trend, annual streamflow, monthly streamflow and daily streamflow, respectively. Overall,
the two models perform similarly. The three missing rate experiments (5%, 10%, and 20%)
perform almost the same as the benchmark (Figures 4 and 5). The coefficient of
determination ($r^2$) between the gap-filled trends and observed trends is more than 0.98 for the
three experiments and two hydrological models.
Since errors in gap-filled trends likely to be different and different time steps when daily
infilled streamflow data is used, we further investigate how gap-filled errors are propagated
from daily to monthly and to annual scales under the three gap-filling cases (5%, 10%, and





20%) (Figures 4 and 5). It is expected that daily gap-filled streamflow has a larger standard
deviation from the benchmark than monthly and annual streamflow since the streamflow was
gap-filled at daily scale. This indicates that the temporal aggregation smooths the gap-filled
error strongly, and it generates very reasonable monthly and annual streamflow estimates
with less standard deviation. It is interesting to note that both models tend to underestimate
very high flows though they are calibrated against the NSE of daily streamflow which puts a
larger weight on correctly representing higher flows.

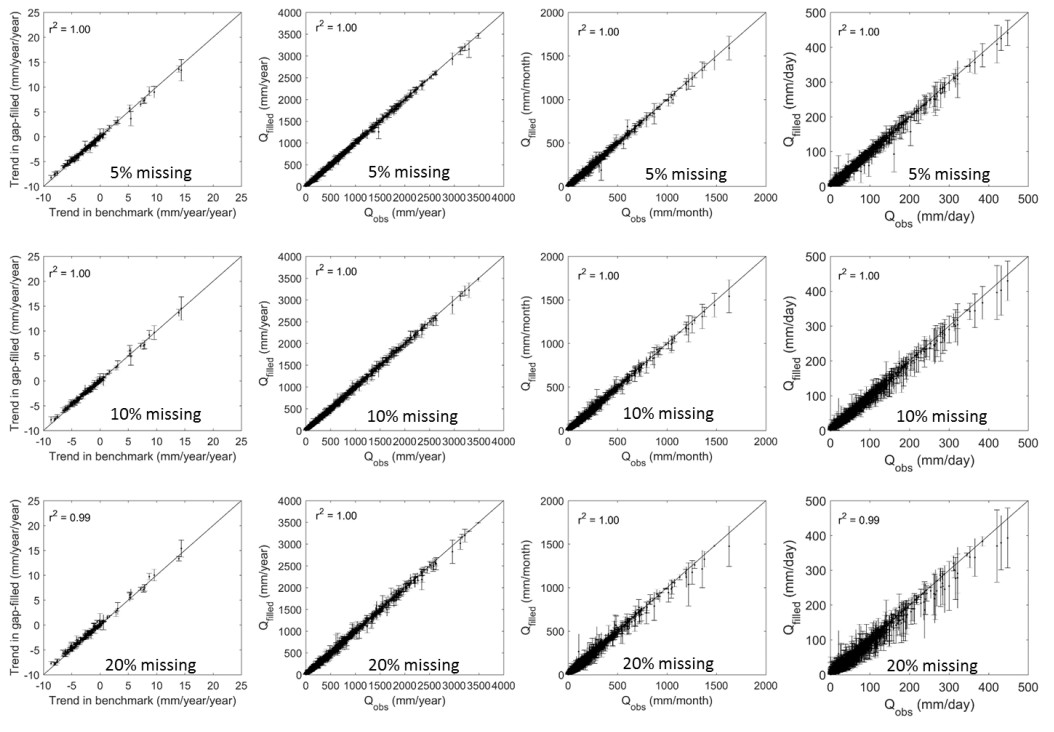

**Fig. 4.** Comparisons between the observed streamflow (x-axis) and gap-filled ones (y-axis) for
streamflow trend (mm/year/year, left panels), annual streamflow (mm/year, second left panels),
monthly streamflow (mm/month, second right panels) and daily streamflow (mm/day, right
panels). The gaps were filled using GR4J. Error bar represents standard deviation of the 100
replicates for each group of 'missing' experiments.

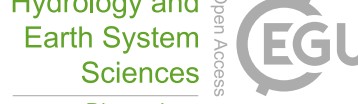



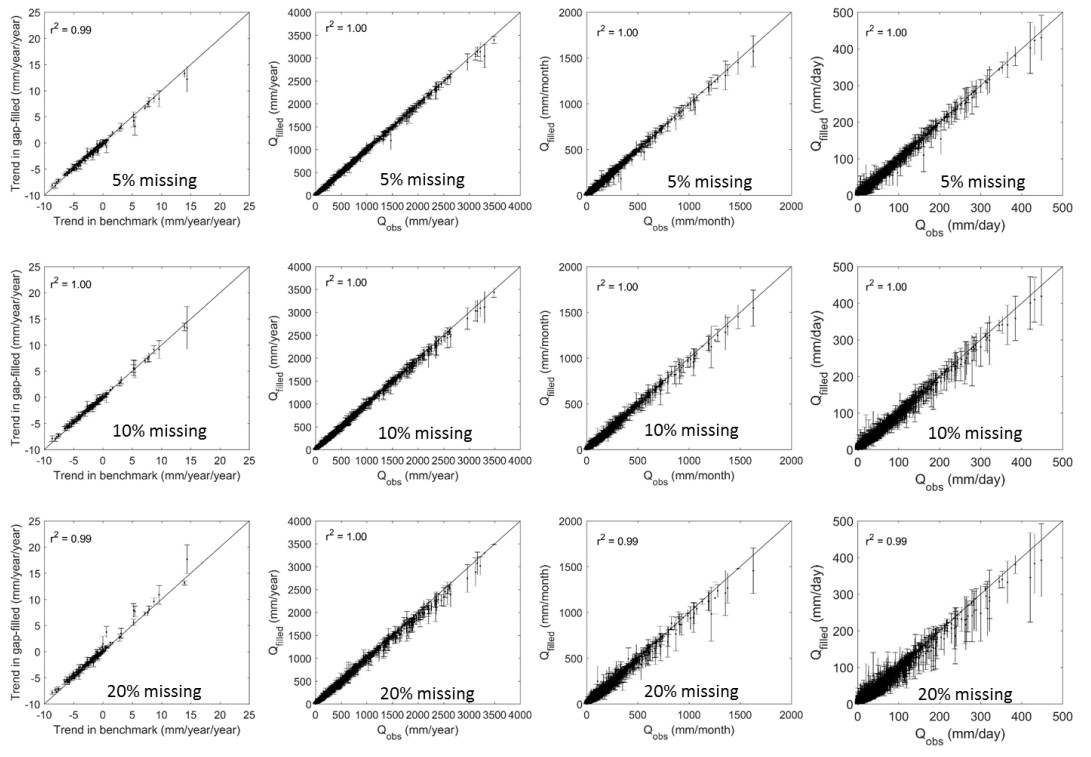

**Fig. 5.** Same as Fig. 4 but using SIMHYD.

Figure 6 further summarises the catchments with trend direction mismatch between the
benchmark and gap-filled data (i.e. change from negative to positive or change from positive
to negative). For the experiments with 5% and 10% missing rates and for GR4J, there are less
than 8 out of the 217 catchments showing a trend mismatch and almost all of them show non-
significant trends ($p > 0.05$). For the experiments with a 20% missing rate for GR4J, there are
less than 10 out of the 217 catchments showing trend mismatch and all of them show non-
significant trends. SIMHYD results are almost the same as GR4J results. All these indicate that
there is very marginal influence on annual streamflow trend directions when the missing rate
is less than 20%.



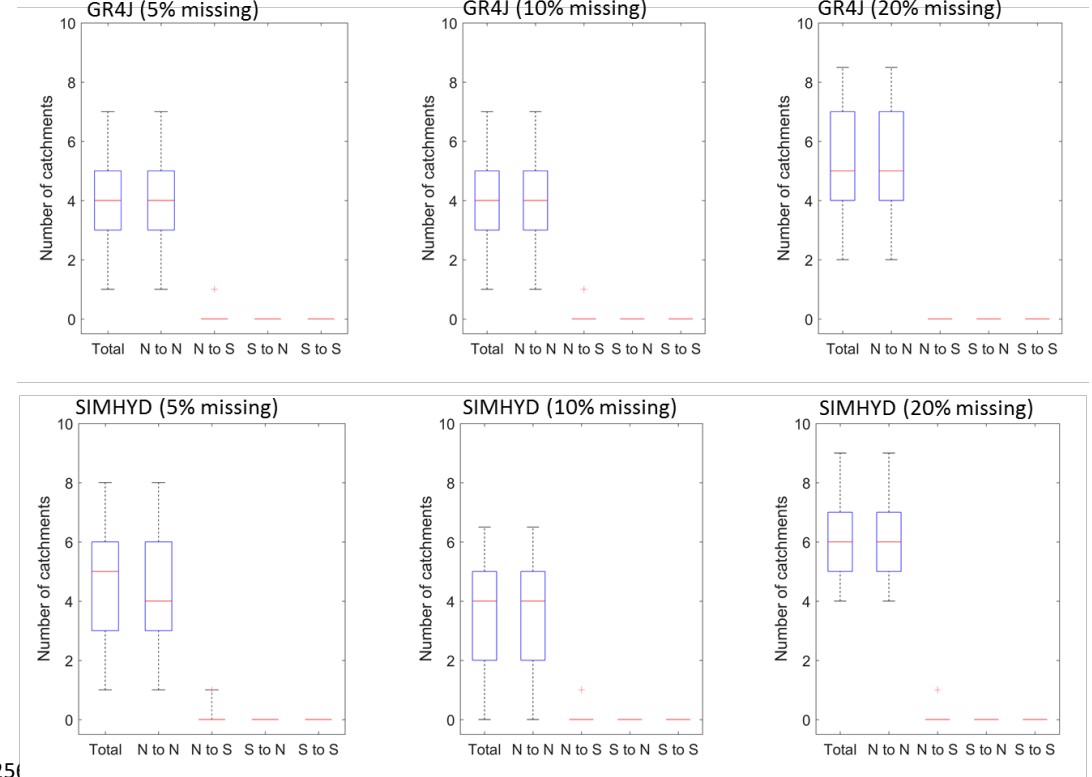

250

**Fig. 6.** Trend mismatch analysis between the gap-filled and benchmark. Total means all

mismatch catchments; 'N' means not significant trends ($p > 0.05$); 'S' means significant

trends ($p \leq 0.05$). The bottom, middle and top of each box are the 25th, 50th and 75th

percentiles, and the bottom and top whiskers are the 5th and 95th percentiles.

Though the three groups of experiments show small trend direction changes (Figure 6), it is

not clear how the trend bias (Eq. 3) looks. To this end, Figure 7 further compares the trend

bias between the experiments. It is clear that the trend biases between 5% and 10% missing

experiments are similar. For GR4J, both have the trend bias varying from -1 to 1

mm/year/year; For SIMHYD, the trend bias between the two is similar when it varies from -

0.5 to 1 mm/year/year, and the trend bias for 5% missing experiment is even larger than that

for 10% missing experiment. The trend bias for 20% missing experiment is noticeably larger



than that for 10% and 5% missing experiments for both models, and the underperformance is
more noticeable from SIMHYD gap-filled than that from GR4J gap-filled.  This result
suggests that the trend bias is reasonable when the missing rate is less than 10%, and can be
large for small number of catchments when the missing rate is to 20%.

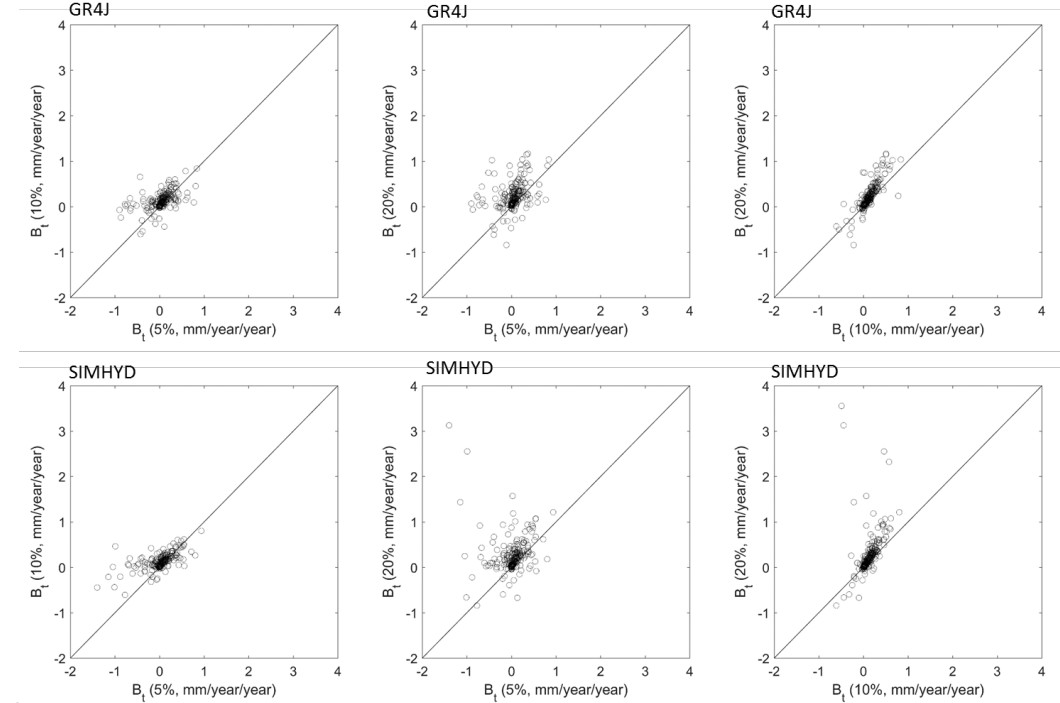


**Fig. 7.** Trend biases comparison between the three groups of gap-filling experiments (5%,
10% and 20%). Top three are for GR4J and bottom three are for SIMHYD.

## 4   Discussion and conclusions

Researchers are keen to have a comprehensive understanding of rules for excluding
catchments with gaps in the streamflow record. Our results indicate that when the streamflow
data gaps are up to 10%, the gap-filled data obtained using hydrological modelling are very
reasonable for annual trend analysis and annual streamflow estimates. Choosing the threshold
of 10% missing rate will allow the use of many more catchments in modelling and data





analysis studies. For example, of the 780 unregulated Australian catchments available for
modelling studies(Zhang et al., 2013), there are 237 catchments with the missing rate of 1-
10% during 1981-2010, accounting for 38% of total available catchments (Figure 1). Of these
237, 67 (~28%) also have gaps lasting more than one year (which we did not consider in this
analysis), and therefore these may not be suitable for use. With an increased number of
catchments, more reliable large-scale hydrological modelling studies can be carried out
(Beck et al., 2016; Parajka et al., 2013; Zhang et al., 2016a).
The 'missing' rate experiments designed in this study are based on the actual data missing
patterns obtained from the 780 catchments. In most cases, the consecutive missing days are
less than 10, as indicated by Figure 3, indicating brief periods of gauge malfunctions. It is
however interesting to note that there are streamflow gaps lasting much longer than this in
many catchments, with gaps of many months in some cases, noting that we excluded gaps
lasting one year or more. It is highly likely that filling a gap of one year or more will result in
biases larger than those presented here.
Furthermore, we also tested the quality of random gap-filled daily streamflow. In that case,
the missing patterns were randomly selected using a random number generator. The results
obtained from the random gap-filling (not shown) are similar to the results presented here.
Thus, it is likely that the length of the gaps (as long as it is less than one year) is unlikely to
impact the results of the gap-filling experiment. We would conclude from this that the use of
hydrologic modelling for filling the substantially gapped data (up to 10% missing rate)
described here for Australia will not impact annual trends of streamflow. Impacts on other
streamflow characteristics also need to be examined, as well as seeing if the results obtained
in Australia are comparable with those in other parts of the world, where the length of
observational gaps may be quite different to those shown in Figure 3.



To understand if the quality of gap-filled streamflow is related to catchment attributes and
calibration accuracy, we conducted further analysis among the trend bias, model calibration
efficiency (i.e. *NSE*) and catchment aridity index (mean annual potential evaporation divided
by mean annual precipitation) (Figure 8). The model calibration results at dry catchments are
normally poorer than those at wet catchments. However, the trend bias (mm/year/year)
obtained from dry catchments is usually smaller. The large biases are observed from the
catchments with aridity index less than 2 and with the calibrated NSE being larger than 0.60.
In part, this is to be expected since the streamflow is also lower in more arid catchments,
meaning that the trend bias is also likely to be lower.
Figure 9 shows the relationship between relative trend bias (%, Eq. 4) and aridity index. It
shows that not only is the actual trend bias lower in drier catchments, but so too is the relative
(%) trend. This result suggests that the large bias in annual trends as a result of gap-filling is
observed in relatively wet catchments where model calibrations are reasonably good. This
result seems counter-intuitive and requires further exploration, which is beyond the scope of
the current paper.





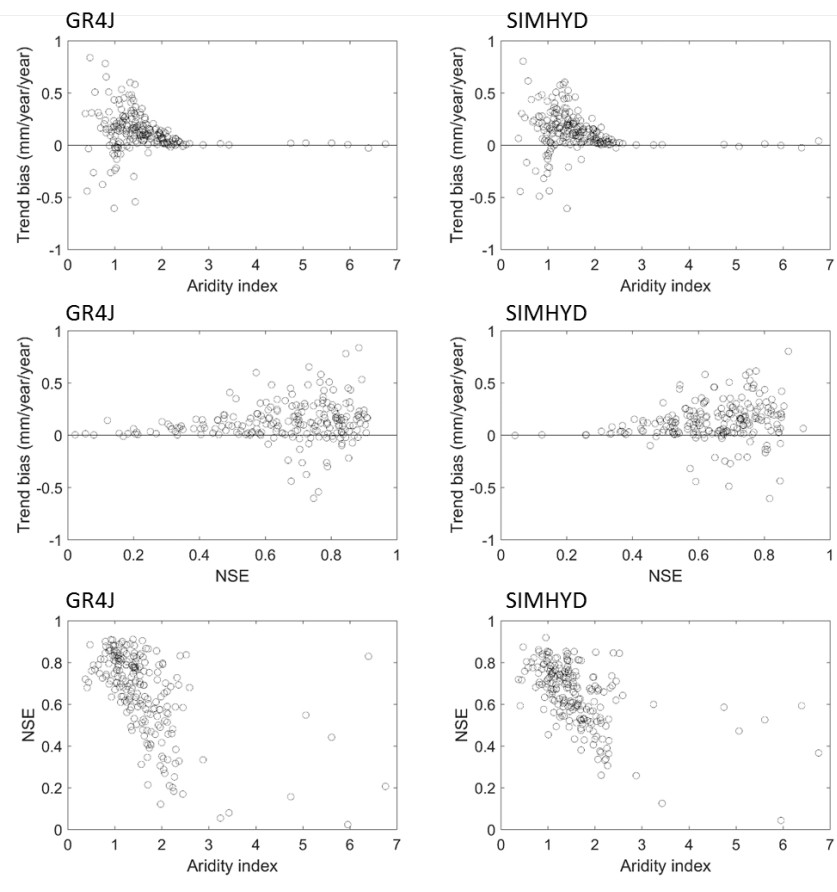


**Fig. 8.** Relationships among trend bias (mm/year/year), model calibration Nash-Sutcliffe

Efficiency and aridity index for each catchment and for the experiment of 10% missing rate.





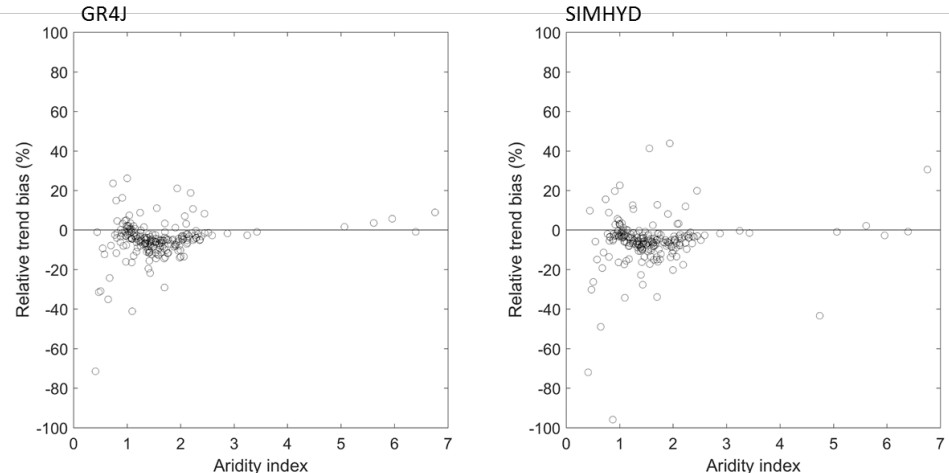


**Fig. 9.** Relationships between relative trend bias (mm/year/year) and aridity index for each

catchment and for the experiment of 10% missing rate.

This study focuses on evaluating annual streamflow and its trends. Therefore, we used the
Nash-Sutcliffe Efficiency plus model bias (Eqs. 5 and 6) to calibrate the two hydrological
models. If other hydrological response variables such as low flow metrics are required, other
model calibration schemes should be used since the NSE model calibration scheme gives
more weight to reproducing high flows at the expense of low-flows (Zhang et al., 2014). Low
flow metrics have important ecological implications(Mackay et al., 2014; Smakhtin, 2001).
In general however, it is challenging to use hydrological modelling for low flow simulations
and predictions(Pushpalatha et al., 2012; Staudinger et al., 2011). To have credible low flow
gap-filling, model calibrations should use an objective function that puts more weights on
low flows, such as NSE of daily inverse streamflow and the direct low flow metrics. Another
possible method is to combine hydrological modelling with other methods for gap-filling,
such as using nearby gauges(Lopes et al., 2016) and statistical methods(Gedney et al.,
2006b).





It is noted that the infilled data purely refers to the 'missing' data. All streamflow gauges are
only rated to a certain flow. Once the flow exceeds that level during flooding, the results are
interpolated using stage-discharge relationships(Peña-Arancibia et al., 2015). These
interpolations could be a major source of observation error. However, investigating high flow
interpolation and data quality is beyond the scope of this study.
In summary, our results clearly demonstrate that the gap-filled data is most accurate when
examining trends at the annual scale, followed by monthly scale, and with least satisfaction at
the daily scale. This gives researchers confidence for annual trend analysis, a hot topic in
hydrological and climate sciences. Our results also clearly indicate that the gap-filling of
Australian streamflow data using hydrological model is very reasonable when the missing
rate is less than 10%, with only a small number of catchments showing a large trend bias
when the missing rate is to 20%. The results also indicate that gap-filling drier catchments
appears to be more successful than gap-filling wetter catchments.
*Acknowledgements.* This study was supported by the CSIRO strategic project "Next
generation methods and capability for multi-scale cumulative impact assessment". Data used
for gap-filling experiments are freely available upon request from the corresponding author
(Yongqiang Zhang, email: yongqiang.zhang2014@gmail.com). The authors declare no
conflict of interests.

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
