# Peer review of "How good are hydrological models for gap-filling"

_Hydrology and Earth System Sciences, 2018_

## Referee Comment (RC1) · M. Zaidman (Referee) · 9 Jul 2018

Overall comments:

Generally a clear well-written paper. The underlying science appears to have been undertaken robustly, methodically and consistently. My main thoughts, having digested the submission, were to the wider scientific significance of the work presented. Has this been suitably explored within the context of the work? Currently the paper has a colloquial emphasis (Australia) and as a reader in the UK, I would like the authors to make a comment on whether the results are transferable elsewhere and also on how much dependency there is on the type of model used for infilling and patterns of missing data. Even for a more direct audience (e.g. users of Australian streamflow data

[Figure]

Creative Commons BY license logo

/ those wising to understand the reliability of trend detection analysis in an Australian context) the benefits/implications of the outcomes of the work could be drawn out in the paper a little more. At the very end of the paper the authors tantalise the reader by hinting at other patterns within the dataset, beyond the scope of the study to explore at this point. Ideally for me, this paper would give more value if it took a stance of saying, having established that gap filling does not impact on trend analysis, what the trend analysis on the gap-filled data shows and whether this changes our perception on the strength and direction of trend for either individual sites or regionally. Finally, what a shame the paper does not address the potential payback of infilling with modelled data compared with other methods (like interpolation or correlations with nearby sites for example). Would there be less confidence in the trend analysis results if modelling had not been used as the gap filling method. Having said the above, I would not object to publication of this paper in its current form (no suggested corrections to the text). It is a self-contained work that no doubt many hydrologists will find useful.

Specific comments:

Abstract: The point that springs to my mind is that if gap filling has so little impact, then why bother to undertake it in the first place? Presumably the gap filling is being undertaken de rigour/as part of data QA for reasons of consistency / completeness and the purpose here is to show this does not have negative impact on key hydrological analyses (of which trend analysis might be just one?). The abstract also states there is a lack of quantitative analysis of gap filled data. Is this really true, across the entirety of the international body of scientific literature.

Data and methods: I'm interested to know whether the timing of missing data impacts on the trend analysis outcomes. Presumably the % rates are across the period of record of each site? Could you explain reasons for the gaps in the records, e.g. are all the stations gauged in the same way or are some types of station/ river more vulnerable to gaps than others (e.g. stations on smaller flashy rivers). Was there consideration of data quality outside of periods with gaps. Are stations with more gaps likely to suffer

poorer data quality overall.

Results: It is stated that the model performance is not as good for high flows, but the analysis considers annual trends (annual average flows?). Was any analysis of trends in high flow patterns attempted and if so was there a different outcome. I'd also like to see more exploration and explanation of differences seen between the SIMHYD and GR4J results. Does one model theoretically out perform the other? Are the differences between the infilled trend analysis for the two models the same order of magnitude as between trend from filled and unfilled series etc. I just wonder if we need more discussion in this section to draw out some useful implications or provisos. Should one model be preferred or give a greater payback (i.e. Gap filling is just as good but the model is more practicable to use/more straight forward to parameterise).

---

## Referee Comment (RC2) · J. Parajka (Referee) · 21 Jul 2018

**General comments**

This study explores the efficiency of gap-filling of streamflow data by using simulations of a hydrologic model. The main objective is to evaluate the annual trends and annual variables obtained from gap-filled streamflow data using two hydrological models (GR4J and SIMHYD) in 217 catchments in Australia. The results show that when the missing rate of streamflow data is less than 10%, the gap-filled streamflow data from hydrological models perform very close to the benchmark data. Interestingly, the relative streamflow trend bias caused by the gap-filling is not very large even in very dry catchments where typically the hydrological model calibration is poor. Authors con-

clude that the gap filling using hydrological modelling has little impact on the estimation of annual streamflow and its trends in selected catchments in Australia.

Overall, the study is very clearly written, has a good structure and it is within the scope of HESS. The presentation of take home messages is very compact and clear. I have only one question which remained unanswered after reading the manuscript. What is the impact of patterns of missing data in terms of dominant hydrologic regime in the catchments? I expect that the large dataset in Australia covers catchments with different hydrological (seasonal) runoff regime. Are the missing data more-less evenly distributed thorough the year in all catchments or are there some seasonal patterns of gaps? What is the impact if majority of missing data are from the most/least important season (in terms of maximum monthly runoff)? I would expect that if the majority of e.g. 10% missing data are from seasons with minimum monthly runoff then the impact on annual mean or trend will be smaller and vice versa. Are there some differences between catchments with different seasonal regime? Some more discussion around it will be interesting.

Finally I would like to congratulate the authors for a very nice analysis. I enjoyed reading it.

---

## Author Comment (AC2) · 22 Jul 2018

**parajka@hydro.tuwien.ac.at**

Overall comments:
General comments
This study explores the efficiency of gap-filling of streamflow data by using simulations of a hydrologic model. The main objective is to evaluate the annual trends and annual variables obtained from gap-filled streamflow data using two hydrological models (GR4J and SIMHYD) in 217 catchments in Australia. The results show that when the missing rate of streamflow data is less than 10%, the gap-filled streamflow data from hydrological models perform very close to the benchmark data. Interestingly, the relative streamflow trend bias caused by the gap-filling is not very large even in very dry catchments where typically the hydrological model calibration is poor. Authors conclude that the gap filling using hydrological modelling has little impact on the estimation of annual streamflow and its trends in selected catchments in Australia.

Overall, the study is very clearly written, has a good structure and it is within the scope of HESS. The presentation of take home messages is very compact and clear. I have only one question which remained unanswered after reading the manuscript. What is the impact of patterns of missing data in terms of dominant hydrologic regime in the catchments? I expect that the large dataset in Australia covers catchments with different hydrological (seasonal) runoff regime. Are the missing data more-less evenly distributed thorough the year in all catchments or are there some seasonal patterns of gaps? What is the impact if majority of missing data are from the most/least important season (in terms of maximum monthly runoff)? I would expect that if the majority of e.g. 10% missing data are from seasons with minimum monthly runoff then the impact on annual mean or trend will be smaller and vice versa. Are there some differences between catchments with different seasonal regime? Some more discussion around it will be interesting.
Finally I would like to congratulate the authors for a very nice analysis. I enjoyed reading it.

Response: We do appreciate the favourable comments from Juraj Parajka. Juraj highlights the science quality of this study and quality presentation.

To address the question Juraj raised regarding seasonal pattern of number of the missing days, we future plot a boxplot plot (Figure 4). Yes, the missing data are more-less evenly distributed through different seasons across all 39 catchments (with missing rate of 8% to 12%) within the 10% missing data group. This basically suggests the streamflow is missing randomly through the year. Having

said that, we actually conducted independent modelling experiments (but did not show them in the previous version) to test the consequence if the missing streamflow only occurs in high-flow or low flow seasons in the extreme cases. *In lines 324 to 334 the text now says "Streamflow data gap could only occur in high flow or low flow condition in the extreme case though majority of missing data for the Australian catchments are more or less evenly distributed through the year. We further tested the impact of filling streamflow data in high flow or low flow condition. In that case, the missing patterns were selected using only high flow (>95th percentile) or low flow (less than 50th percentile) data. The results obtained from the low flow gap-filling indicates that there is only a negligible influence on annual streamflow trend estimates when the missing rate of is less than 50%. In contrast, the high flow gap-filled shows a noticeable change in annual streamflow trend when the missing rate is 5% (or at 95th percentile). This is understandable since high flow is usually several orders of magnitude higher than low flow, and a certain error in filling high flow could have large impact on annual flow and its trends".*

[Figure]

Fig. 4. Distribution of number of missing days across different seasons, summarised from 39 catchments with a missing rate ranging from 8% to 12% (i.e. 10% missing data group)

---

## Author Response (AR1)

CSIRO Land and Water
Clunies Ross Street, Black Mountain, Canberra ACT
GPO Box 1700, Canberra ACT 2601 Australia
Telephone: +61 2 6246 5700  Facsimile: +61 2 6246 5800
www.csiro.au
ABN 41 687 119 230

07 August 2018

Dr Louise Slater

Editor for Hydrology and Earth System Sciences

Dear Louise,

First, we would like thank you and the two reviewers for the quite favourable reviews on our manuscript "How good are hydrological models for gap-filling streamflow data?" (hess-2018-250). We appreciate that all of you acknowledge that this is a concise but very interesting paper. It is really encouraging. Although the reviewers provided favourable comments and acknowledge the research value of this paper, they also gave insightful comments to clarify several important points, i.e. seasonality of missing data, model comparison, broad implications. All of these comments have been carefully considered and all comments have been adopted and incorporated to the improved revised version. The follows are key improvements:

   a. New analysis on seasonality of missing data;
   b. New analysis on model comparisons;
   c. More literature review; and
   d. More discussion on the broad implications, comparison for various approaches and future directions.

In the following sections, we provide point-to-point response to the comments, followed by the track changed version. Please let us know if there are any questions. Thanks again for you and the reviewers for your time, suggestions and comments.

Yours sincerely,

Yongqiang Zhang (on behalf of all co-authors)

Principal Research Scientist

CSIRO Land and Water

Clunies Ross St. Canberra ACT 2601, Australia

Dear Yongqiang Zhang,

We are pleased to inform you that the Editor report for the following manuscript is now available:

Journal: HESS

Title: How good are hydrological models for gap-filling streamflow data?

Author(s): Yongqiang Zhang and David Post MS No.: hess-2018-250 MS Type: Research article

Iteration: Minor Revision

The Editor has decided that minor revisions are necessary before the manuscript can be accepted. Please find the Editor Report at https://editor.copernicus.org/HESS/ms_records/hess-2018-250.

We kindly ask you to revise your manuscript accordingly and to upload the revised files, a point-by-point reply to the comments, and a marked-up manuscript version showing the changes made in your File Manager no later than 14 Aug 2018: https://editor.copernicus.org/HESS/file_manager/hess-2018-250. Please find all information on manuscript submission under https://www.hydrology-and-earth-system-sciences.net/for_authors/submit_your_manuscript.html.

Your revised manuscript will be reviewed by the Editor and you will be informed about the outcome by separate email.

Besides adjustments requested by the Editor or Referees, please check your manuscript carefully for typos, missing co-authors and their affiliations, terminology, updates of data in tables, or updates of variables in equations. All these have to be clarified with the Editor and therefore have to be included before you submit your revised manuscript. Should your manuscript be finally accepted it will not be possible to include such rather substantial changes anymore when your manuscript is in final production (proofreading).

To log in, please use your Copernicus Office user ID 217293.

Please note that all Referee and Editor reports, the author's response, as well as the different manuscript versions of the peer-review completion (post-discussion review of revised submission) will be published if your paper will be accepted for final publication in HESS.

You are invited to monitor the processing of your manuscript via your MS Overview: https://editor.copernicus.org/HESS/my_manuscript_overview

In case any questions arise, please contact me. Thank you very much for your cooperation.

Kind regards,

Natascha Töpfer

Copernicus Publications

Editorial Support

editorial@copernicus.org

on behalf of the HESS Editorial Board

Reviewer comments are in black and our responses are provided in this blue colour. We also use codes R1C1 to mean Reviewer 1 Comment 1, to allow for cross-referencing in the response letter and to aid navigation.

COMMENTS FROM EDITORS AND REVIEWERS:

Dear Authors,

Thank you for your responses to the two referees' reports. Based on my own reading of the manuscript, I find this is a concise but interesting paper that fits the scope of HESS well and will be of interest to the community.

EC1): The two reviews are both quite favourable, but they also make some important points about comparing the effect of gap-filling in different types of sites (e.g. in different regions and hydrological regimes), providing contextualisation and implications of the work (its wider significance and transferability to other contexts), and comparing the models. Additionally, some of the discussion and assertions could be better supported by references, such as the statement that 'it is well recognised that hydrological modelling is the best option' (for gap-filling). See for instance papers describing the utility/efficacy of different gap-filling approaches (e.g. those discussed in https://doi.org/10.1002/joc.4954).

I would therefore like to invite you to upload a revised manuscript, incorporating the proposed changes and additions, and making any other modifications where you see fit.

I look forward to receiving the revised manuscript.

With best regards,

Louise Slater

Response:

Thanks for you quite favourable and constructive comments. As stated in the above letter to Editor, we made following revisions based on the comments from Editor and two reviewers Maxine Zaidman and Juraj Parajka:

    a.  New analysis on seasonality of missing data;

b. New analysis on model comparisons;
c. More literature review; and
d. More discussion on the broad implications, comparison for various approaches and future directions.

We add more literatures on various approaches for data gap-filling. In lines 60-63, the clear text now says "*There are many methods used for gap-filling the missing data, including interpolation from nearby gauges (Hannaford and Buy, 2012; Lavers et al, 2010; Lopes et al., 2016), statistical methods (Gedney et al., 2006b), hydrological modelling (Dai et al., 2009; Sanderson et al., 2012), and multiple infilling methods (Harvey et al., 2012).*". We add more references to support the use of hydrological modelling approaches. In lines 63-67, the text now says "*Among them, the hydrological modelling method is widely used since it fully considers the spatial heterogeneity and temporal variability of climate forcing data, and can achieve sufficient simulations when it is calibrated against a small number of observations (Peña-Arancibia et al. 2014; Rojas-Serna et al., 2016; Seibert and Beven, 2009; Liu and Zhang, 2017)*".

We also add discussion on high-flow gap filling impacts. In lines 334-337, the clear text now says "*In contrast, the high flow gap-filled data shows a noticeable change in annual streamflow trend when the missing rate is 5% This is understandable since high flow is usually several orders of magnitude higher than low flow, and errors in filling high flow could have large impacts on annual flow and its trends (Slater and Villarini, 2017).*".

We also discuss more comparisons between hydrological modelling and other gap filling approaches. In lines 388-394, the text now says "*It would also be interesting to compare hydrological modelling to other approaches for filling streamflow data gaps. Hydrological modelling is a most useful method used in Australia for predicting daily streamflow in ungauged catchments (Chiew et al., 2009; Li and Zhang, 2017; Zhang and Chiew, 2009; Viney et al., 2009). It has been used operationally by the Australian Bureau of Meteorology for filling daily streamflow data gap for many years. In the future, this operational method could further be comprehensively evaluated against other approaches, such as interpolation or correlations with nearby gauging sites.*".

New References:

Hannaford, J., and Buys, G.: Trends in seasonal river flow regimes in the UK, Journal of Hydrology, 475, 158-174, 10.1016/j.jhydrol.2012.09.044, 2012.

Harvey, C. L., Dixon, H., and Hannaford, J.: An appraisal of the performance of data-infilling methods for application to daily mean river flow records in the UK, Hydrology Research, 43, 618-636, 10.2166/nh.2012.110, 2012.

Lavers, D., Prudhomme, C., and Hannah, D. M.: Large-scale climate, precipitation and British river flows Identifying hydroclimatological connections and dynamics, Journal of Hydrology, 395, 242-255, 10.1016/j.jhydrol.2010.10.036, 2010.

Sanderson, M. G., Wiltshire, A. J., and Betts, R. A.: Projected changes in water availability in the United Kingdom, Water Resources Research, 48, 10.1029/2012wr011881, 2012.

Slater, L., and Villarini, G.: On the impact of gaps on trend detection in extreme streamflow time series, International Journal of Climatology, 37, 3976-3983, 10.1002/joc.4954, 2017.
* * *
Referee #1

**M. Zaidman (Referee)**
maxine.zaidman@ jbaconsulting.com

Overall comments:
R1C1): Generally a clear well-written paper. The underlying science appears to have been undertaken robustly, methodically and consistently. My main thoughts, having digested the submission, were to the wider scientific significance of the work presented. Has this been suitably explored within the context of the work? Currently the paper has a colloquial emphasis (Australia) and as a reader in the UK, I would like the authors to make a comment on whether the results are transferable elsewhere and also on how much dependency there is on the type of model used for infilling and patterns of missing data. Even for a more direct audience (e.g. users of Australian streamflow data / those wising to understand the reliability of trend detection analysis in an Australian context) the benefits/implications of the outcomes of the work could be drawn out in the paper a little more. At the very end of the paper the authors tantalise the reader by hinting at other patterns within the dataset, beyond the scope of the study to explore at this point. Ideally for me, this paper would give more value if it took a stance of saying, having established that gap filling does not impact on trend analysis, what the trend analysis on the gap-filled data shows and whether this changes our perception on the strength and direction of trend for either individual sites or regionally. Finally, what a shame the paper does not address the potential payback of infilling with modelled data compared with other methods (like interpolation or correlations with nearby sites for example). Would there be less confidence in the trend analysis results if modelling had not been used as the gap filling method. Having said the above, I would not object to publication of this paper in its current form (no suggested corrections to the text). It is a self-contained work that no doubt many hydrologists will find useful.

Response: We do appreciate the favourable comments from Maxine Zaidman. Maxine highlights the underlying science appears to have been undertaken robustly, methodically and consistently.

We are grateful for her thoughtful thinking on how to transfer the results obtained from Australia to other parts of the work. It is indeed it is important to discuss the implication. To this end, we add one paragraph in Discussion section. In lines 377-387, the text says "*The modelling experiments and findings from this study could have important implications for*

*other parts of the world as well as Australia. First, to develop appropriate gap-filling modelling experiments, it is necessary to evaluate the distribution of consecutive missing data pattern. The probability distribution of consecutive missing data is skewed toward the low end, which can be nicely simulated using the Gamma distribution (Eq.1). This distribution should be very useful for similar missing patterns in other regions. Second, hydrological modelling is a very good tool for filling gaps since it can fully take the advantage of climate forcing and non-gap streamflow data, and obtain the best possible daily simulations. Third, the threshold of 10% identified in this study should be applicable to regions/catchments with similar missing patterns. However, if the data gaps continue for seasons or years, the threshold may not hold."*

In terms of comparisons between modelling and other methods (like interpolation or correlations with nearby sites for example), it is well recognised that hydrological modelling in Australia is the best option since it fully takes advantage for climate forcing and non-gap streamflow data. We add one paragraph for discussing the comparison. In lines 388 to 394, the text now says "*It would also be interesting to compare hydrological modelling to other approaches for filling streamflow data gaps. Hydrological modelling is a most useful method used in Australia for predicting daily streamflow in ungauged catchments (Chiew et al., 2009; Li and Zhang, 2017; Zhang and Chiew, 2009; Viney et al., 2009). It has been used operationally by the Australian Bureau of Meteorology for filling daily streamflow data gap for many years. In the future, this operational method could further be comprehensively evaluated against other approaches, such as interpolation or correlations with nearby gauging sites*".

Specific comments:
R1C2) Abstract: The point that springs to my mind is that if gap filling has so little impact, then why bother to undertake it in the first place? Presumably the gap filling is being undertaken de rigour/as part of data QA for reasons of consistency / completeness and the purpose here is to show this does not have negative impact on key hydrological analyses (of which trend analysis might be just one?). The abstract also states there is a lack of quantitative analysis of gap filled data. Is this really true, across the entirety of the international body of scientific literature.

Response: In our knowledge, it is indeed that there is lack of quantitative evaluation of the gap-filled data accuracy in most hydrological studies. The scientists basically use a threshold, based on some kind of gut feeling. This study can fill knowledge gap. This study provides two key findings: (1) when the missing rate is less than 10%, the gap-filled streamflow data obtained using calibrated hydrological models perform almost as same as the benchmark data (less than 1% missing) for estimating annual trends for 217 unregulated catchments widely spread in Australia; (2) the relative streamflow trend bias caused by the gap-filling is not very large in very dry catchments where the hydrological model calibration is normally poor. In terms of why it is undertaken in the first place, it is generally done by the collecting agency (in Australia, the Bureau of Meteorology), as end users often require streamflow data with no gaps.

R1C3. Data and methods: I'm interested to know whether the timing of missing data impacts on the trend analysis outcomes. Presumably the % rates are across the period of record of each site? Could you explain reasons for the gaps in the records, e.g. are all the stations gauged in the same way or are some types of station/ river more vulnerable to gaps than

others (e.g. stations on smaller flashy rivers). Was there consideration of data quality outside of periods with gaps. Are stations with more gaps likely to suffer poorer data quality overall.

Response: In most cases, missing data are randomly distributed and different gauges show different missing pattern. This can be seen from missing patterns in Fig. 3 that there is a skewed distribution for consecutive missing days. This means that majority of the consecutive missing days are less than 30 days. The data gaps for Australian streamflow gauges mainly include (see lines 95-97):
1. Non-sensible record
2. Sensor broken
3. No recorded data (Instrumentation removed)
4. No data exists
5. No record or record lost

In terms of timing of missing data and reasons of gaps, we further plot a boxplot plot (Figure 4). Yes, the missing data are more-less evenly distributed through different seasons across all 39 catchments (with missing rate of 8% to 12%) within the 10% missing data group. This indicates that the data gaps were not skewed toward a particular season and it occurred randomly through the year. Having said that, we actually conducted independent modelling experiments (but did not show them in the previous version) to test the consequence if the missing streamflow only occurs in high-flow or low flow seasons in the extreme cases. *In lines 327 to 337, the text now says "It is possible that data gaps may only exist during high flow or low flow conditions, although that is not what we observed here with the majority of missing data being more or less evenly distributed throughout the year (Figure 4). We did however test the impact of filling streamflow data in high flow or low flow conditions (results not shown here). In those cases, the missing patterns were selected using only high flow (>95th percentile) or low flow (less than 50th percentile) data. The results obtained from the low flow gap-filling indicates that there is only a negligible influence on annual streamflow trend estimates when the missing rate is less than 50%. In contrast, the high flow gap-filled data shows a noticeable change in annual streamflow trend when the missing rate is 5% This is understandable since high flow is usually several orders of magnitude higher than low flow, and errors in filling high flow could have large impacts on annual flow and its trends (Slater and Villarini, 2017)."*.

[Figure]

Fig. 4. Distribution of number of missing days across different seasons, summarised from 39 catchments with a missing rate ranging from 8% to 12% (i.e. 10% missing data group).

We did not consider data quality outside of periods with gaps.

In term of "Are stations with more gaps likely to suffer poorer data quality overall", we do not sure what the poor data quality refer to. If the review talked about poorer simulation quality, we compared the trends between the three gap-filling experiments (Fig. 7). It is clear that the trend biases between 5% and 10% missing experiments are similar. For GR4J, both have the trend bias varying from -1 to 1 mm/year/year; For SIMHYD, the trend bias between the two is similar when it varies from -0.5 to 1 mm/year/year, and the trend bias for 5% missing experiment is even larger than that for 10% missing experiment. The trend bias for 20% missing experiment is noticeably larger than that for 10% and 5% missing experiments for both models, and the underperformance is more noticeable from SIMHYD gap-filled than that from GR4J gap-filled. This result suggests that the trend bias is reasonable when the missing rate is less than 10%, and can be large for small number of catchments when the missing rate is to 20%.

R1C4. Results: It is stated that the model performance is not as good for high flows, but the analysis considers annual trends (annual average flows?). Was any analysis of trends in high flow patterns attempted and if so was there a different outcome. I'd also like to see more exploration and explanation of differences seen between the SIMHYD and GR4J results. Does one model theoretically outperform the other? Are the differences between the infilled trend analysis for the two models the same order of magnitude as between trend from filled and unfilled series etc. I just wonder if we need more discussion in this section to draw out some useful implications or provisos. Should one model be preferred or give a greater

payback (i.e. Gap filling is just as good but the model is more practicable to use/more straight forward to parameterise).

*Response: The two model are overall good for high flow simulations as demonstrated by high NSE and low bias. It only slightly underestimates very high flow (i.e. floods).*

*We have not had analysis of trends in high flow patterns.*

*We include a comparison between SIMHYD and GR4J models. Figure 5 summarises the Comparisons between calibrated GR4J and calibrated SIMHYD for 44 catchments of the 5% missing experiment, 39 catchments of the 10% missing experiment, and 22 catchments of the 20% missing experiment. It is in general that there is no systematic difference between the two. In lines 232-237, the text now says "Overall, the two models perform well and GR4J does not systematically outperform SIMHYD (Figure 5). For the three groups of gap-filling experiments, these two models performs similarly (i.e. the difference of NSE of daily runoff between two is less than 0.02) in 18-19% catchments; SIMHYD model outperforms GR4J model (NSE difference between two is larger than 0.02) in 30-31% catchments; GR4J model outperforms SIMHYD model in 50-51% catchments". In 18-19% catchments, these two models performs similarly (i.e. the difference of NSE of daily runoff between two is less than 0.02); in 30-31% catchments SIMHYD model outperforms GR4J model (NSE difference between two is larger than 0.02); in 50-51% catchments, GR4J model outperforms SIMHYD model.*

[Figure]

*Fig. 5. Comparisons between calibrated GR4J and calibrated SIMHYD for 44 catchments of the 5% missing experiment, 39 catchments of the 10% missing experiment, and 22 catchments of the 20% missing experiment. In each catchment, there were 100 replicates carried out.*

*We also compare the difference between the infilled trends for the two models to the difference the infilled and infilled trends. As shown in the following figure, they are with the similar order (but this figure is not shown in the main text).*

[Figure]

[Figure]
* * *
Referee #2

**J. Parajka (Referee)**
**parajka@hydro.tuwien.ac.at**

Overall comments:
General comments
R2C1. This study explores the efficiency of gap-filling of streamflow data by using
simulations of a hydrologic model. The main objective is to evaluate the annual trends and
annual variables obtained from gap-filled streamflow data using two hydrological models
(GR4J and SIMHYD) in 217 catchments in Australia. The results show that when the missing
rate of streamflow data is less than 10%, the gap-filled streamflow data from hydrological
models perform very close to the benchmark data. Interestingly, the relative streamflow trend
bias caused by the gap-filling is not very large even in very dry catchments where typically
the hydrological model calibration is poor. Authors conclude that the gap filling using
hydrological modelling has little impact on the estimation of annual streamflow and its trends
in selected catchments in Australia.

Overall, the study is very clearly written, has a good structure and it is within the scope of HESS. The presentation of take home messages is very compact and clear. I have only one question which remained unanswered after reading the manuscript. What is the impact of patterns of missing data in terms of dominant hydrologic regime in the catchments? I expect that the large dataset in Australia covers catchments with different hydrological (seasonal) runoff regime. Are the missing data more-less evenly distributed thorough the year in all catchments or are there some seasonal patterns of gaps? What is the impact if majority of missing data are from the most/least important season (in terms of maximum monthly runoff)? I would expect that if the majority of e.g. 10% missing data are from seasons with minimum monthly runoff then the impact
on annual mean or trend will be smaller and vice versa. Are there some differences between catchments with different seasonal regime? Some more discussion around it will be interesting.
Finally I would like to congratulate the authors for a very nice analysis. I enjoyed reading it.

Response: We do appreciate the favourable comments from Juraj Parajka. Juraj highlights the science quality of this study and quality presentation.

To address the question Juraj raised regarding seasonal pattern of number of the missing days, we have included a new boxplot in the paper (Figure 4). Yes, the missing data are more-less evenly distributed through different seasons across all 39 catchments (with missing rate of 8% to 12%) within the 10% missing data group. This basically suggests the streamflow is missing randomly through the year. Having said that, we actually conducted independent modelling experiments (but did not show them in the previous version) to test the consequence if the missing streamflow only occurs in high-flow or low flow seasons in the extreme cases. *In lines 327 to 337 the text now says "
[revised manuscript text omitted]